# A Multimodal Feature Fusion Framework for Sleep-Deprived Fatigue Detection to Prevent Accidents

**DOI:** 10.3390/s23084129

**Published:** 2023-04-20

**Authors:** Jitender Singh Virk, Mandeep Singh, Mandeep Singh, Usha Panjwani, Koushik Ray

**Affiliations:** 1EIE Department, Thapar Institute of Engineering and Technology, Patiala 147001, India; 2DIPAS, Defence Research and Development Organisation, Delhi 110054, India

**Keywords:** alertness, fatigue, non-intrusive, sleep deprivation, voice analysis

## Abstract

Sleep-deprived fatigued person is likely to commit more errors that may even prove to be fatal. Thus, it is necessary to recognize this fatigue. The novelty of the proposed research work for the detection of this fatigue is that it is nonintrusive and based on multimodal feature fusion. In the proposed methodology, fatigue is detected by obtaining features from four domains: visual images, thermal images, keystroke dynamics, and voice features. In the proposed methodology, the samples of a volunteer (subject) are obtained from all four domains for feature extraction, and empirical weights are assigned to the four different domains. Young, healthy volunteers (*n* = 60) between the age group of 20 to 30 years participated in the experimental study. Further, they abstained from the consumption of alcohol, caffeine, or other drugs impacting their sleep pattern during the study. Through this multimodal technique, appropriate weights are given to the features obtained from the four domains. The results are compared with k-nearest neighbors (kNN), support vector machines (SVM), random tree, random forest, and multilayer perceptron classifiers. The proposed nonintrusive technique has obtained an average detection accuracy of 93.33% in 3-fold cross-validation.

## 1. Introduction

Sleep deprivation has a considerable impact on human motor function and cognitive impairments. Lack of sleep results in the reduction of a person’s ability to perform a variety of psychomotor tasks by increasing reaction times for simple and complex tasks. For example, sleep deprivation has been associated with longer reaction times and reduced force [1]. Impairment of alertness poses a danger not only to an individual but also often to the public at large. Fatigue caused by prolonged sleepiness is the predominant risk factor in driving and is probably the most critical one. The term fatigue may seem analogous to sleepiness but rather has a different meaning. Fatigue is defined as the collective physiological and psychological disinclination to perform a task; it arises from prolonged physical or emotional work engagement or from monotony due to the frequency of work engagement [2]. Numerous researchers have reported that prolonged sleep deprivation results in a decline in cognitive abilities. Henceforth, sleep deprivation results in fatigue and increases the possibility of human error, which further results in fatal accidents [3]. Therefore, assessment of the fatigue induced due to sleep deprivation is an important task.

Various physiological features like blood volume pulse (BVP), electroencephalography (EEG), electromyography (EMG), and electrooculography (EOG) are intrusive means of detection, as many electrodes must be put on the subject’s body to acquire the signal [4]. The intrusive methods have some inherent disadvantages, like a biasing in the received signal due to the subject’s awareness of being examined. Recently researchers have successfully implemented nonintrusive methods, like image analysis (eye-blinking rate, pupil diameter), voice analysis (speech patterns), and body reflex analysis, to assess the fatigue induced by the reduced sleeping time [2,4]. However, there is a scope to improve the accuracy of detection methods. 

The present research work proposes improving sleep deprivation-induced fatigue detection by combining multiple domain information acquired in a nonintrusive manner. Subsequently, a computer-based multimodal feature fusion system has been developed for fatigue detection, in which the ambulatory subject is not aware of being examined.

### 1.1. Literature Review

In past research work, various intrusive and nonintrusive methods have been implemented for detecting fatigue. Since this research work has been carried out for multi-domain-based sleep-deprive-induced fatigue detection; thus, we considered intrusive and nonintrusive techniques and recent research considering all four domains for the literature review section.

#### 1.1.1. Literature Review Based upon Facial Features

Fatigue detection through facial features is an active research area, and several techniques are being developed to detect fatigue through facial features, including eye-tracking for eye movements, such as longer blinks. Fatigue can be detected by analyzing facial features, such as drooping eyelids, redness in the eyes, and changes in facial expressions. K. A. Brookhuis et al. discussed traffic accidents caused by inadequate mental workload, i.e., low vigilance or elevated stress. Further, they exhibited the possibility of modern high-end driving simulators for monitoring mental workload while performing a task [5]. Vural et al. employ a machine learning technique for determining the influence of drowsiness on human behavior. The system can approximately detect sleepiness with 90% accuracy [6]. Similarly, an approach for drowsiness detection from greyscale images was suggested by M. J. Flores et al. in 2008. They considered the eye-blinking rate as an indicator to detect a driver’s fatigue [7]. X. Fan et al. conceptualized monitoring human fatigue from facial image sequences by deploying the Gabor-based dynamic representation. In their approach, the image sequence is segregated into dynamic units, and further, a histogram of each dynamic unit is combined as a dynamic feature [8]. In 2013, N. Sharma et al. used thermal imaging of facial regions. Their work is based on the facial features and thermal features acquired through video input. Additionally, they developed a database named ANU Stress DB. However, they obtained a moderately accurate system with 72% detection accuracy through feature fusion [9]. 

#### 1.1.2. Literature Review Based upon Vocal Features

Human voice features can be considered for detecting fatigue due to sleep deprivation. Vocal features include speech analysis to detect the changes in pitch, tone, and rate of speech, which are significant indicators of fatigue. The voice of a fatigued person tends to become more monotone and less expressive. The human voice is also related to a person’s drowsiness and alertness; therefore, it is considered for the study. Hansen et al. performed experimental work to analyze the voice under stress. They explored various stress perception characteristics that affect the speech production system [10]. In comparison, Tawari et al. introduced a feature set based on the cepstrum analysis of voice pitch and voice intensity. An overall accuracy of around 84% is reported by them [11]. R. Fernandez et al. demonstrate a speech recognition model to evaluate the structure of naturally occurring speech of various subjects. They reported 70% detection accuracy and 30% false alarms [12]. Mel-frequency cepstral coefficient (WMFCC) features are considered for spectral weights from a recorded speech sample by E. Bozkurt et al. They concluded that the performance of classifiers based on WMFCC features is better than those with conventional spectral features [13]. J. Krajewski et al. evaluated various classifiers, and the phonetic feature set models’ vocal features for fatigue detection achieved 78.3% accuracy for speaker classification [14]. For fatigue detection, M.J. Caraty et al. experimented with continuous oral reading patterns of various subjects. They implemented a dual-class SVM classifier and achieved an accuracy of 94.1% for the training set and 68.2% for the test set [15]. Body reflex is an unplanned involuntary action in response to a stimulus. Sleep deprivation also causes muscle fatigue, affecting body reflexes like keyboard typing and hand grips. A. Jaimes et al. studied the methods focused on the body, gesture, and eye gaze, including emotion analysis from audio [16]. 

#### 1.1.3. Literature Review Based upon Reflex Analysis

Reflex analysis is another way to detect fatigue by measuring reaction time. Fatigue contributes to slowing down the reflexes of a person, resulting in elongated reaction time. Analyzing a reaction time is straightforward, such as pressing a button in response to a visual stimulus, which can be used to measure reaction time. It is a promising approach having potential in research studies. L. M. Vizer et al. proposed a methodology to detect stressed subjects by monitoring the interaction pattern of a keyboard. Outcomes from their experimental study exhibit the possibility for stress classification through keystroking patterns [17]. Y. M. Lim et al. investigated the characteristics of keyboard and mouse activity to classify stressed subjects. They concluded that the keyboard and mouse behavior is affected due to the stress induced by time pressure and mental arithmetic problem [18]. In addition to this, Nahin et al. discussed an approach to detect a state of mind by analyzing keyboard typing patterns. The collective keystroke dynamics and text sequence analysis show 80% accuracy [19]. Hooda R et al. studied the necessity of real-time fatigue detection more accurately, including biological and physical features [20]. 

#### 1.1.4. Recent Literature Review Based upon Modern Techniques

Overall, detecting fatigue is a complex task, and it often requires a combination of techniques to achieve accurate results. However, with the improvement in modern technologies and machine learning algorithms, detecting fatigue through multimodal features is becoming easier. V. J. Kartsch et al. studied the various research efforts made to compute the degree of drowsiness by eye-tracking and analyzing physiological features through EEG signals [21]. Similarly, Chunhua et al. assessed mental fatigue through EEG to prevent the risk of performance degradation. The accuracy of their language understanding experimental approach is 87.9% [22]. N. Wu et al. experimented on the speech signal by extracting the Mel frequency cepstrum coefficient (MFCC) and implementing machine learning techniques for estimating fatigue. They compared various algorithms, including the self-adaption quantum genetic algorithm (SQGA), back propagation neural network (BPNN), k-nearest neighbor (kNN), and support vector machine (SVM). 94.0% detection accuracy was obtained only by SVM [23]. Similarly, AI Siam et al. experimented with emotion recognition through facial images and speech. Their real-time deep learning approach included machine vision and speech modalities. Furthermore, they used principal component analysis (PCA) for feature decomposition. They claimed 97% detection accuracy for simulation only. [24]. A. A. Alnuaim proposed a real-time approach to emotion detection through speech to develop modules for speech emotion recognition (SER) to support applications related to human-computer interfaces (HCI). In their research, they discussed the results of the various classification approaches and feature fusion to enhance emotion detection accuracy. Unlike most previous studies, their analysis is based on multi-lingual data sets [25]. T. Tuncer et al. proposed a framework based on EEG to detect driver fatigue. In their approach, they presented a hybrid three-layered feature selection method. Their nonintrusive method obtained 97.29% classification accuracy through EEG signals for driver fatigue detection [26]. B. Fatima et al. investigated micro-sleep patterns and developed a cost-effective solution for detecting driver fatigue. They distinguished open and closed eyes through SVM and AdaBoost algorithms. Micro-sleep patterns are determined to detect fatigue state and eventually trigger a warning alarm. They achieved an average accuracy of 96.5% and 95.4% for SVM and AdaBoost, respectively [27]. In recent research, K. O’Keeffe et al. investigated the effect of fatigue on human performance and investigated the efficacy of the existing methods for inducing mental fatigue. In their experiment, they conducted six sessions with 12 participants and performed two cognitive tests; the first was an AX-continuous performance test (AX-CPT), and the second was the TloadDback test [28]. K.J. Heaton et al. worked on the prediction of the changes the performance through electrodermal activity and speech-motor coordination. They interpreted the decline in cognitive performance and alertness with the increase in workload [29]. G. Sikander et al. studied the various techniques for detecting driver fatigue. Nowadays, automakers are installing driver assistance technologies (ADAS) in automobiles which include fatigue detection devices. They reviewed and compared numerous approaches and methodologies for the detection of fatigue through physiological features and vehicular features [30]. M. Doudou et al. proposed an embedded methodology to monitor drivers’ alertness and detect real-time drowsiness levels. Their research is based on vehicle behavior and driver behavior through video and physiological signals to figure out the applicability and accuracy in terms of intrusiveness and practical usage viewpoint [4]. In recent research, C. Zhao emphasized driving safety in the modern era of smart devices and the proliferation of the internet of vehicles (IoV). They implemented a Bayesian convolutional neural network (BCNN) data selection strategy and introduced the uncertainty-weighted asynchronous aggregation (UWAA) algorithm [31]. Moreover, Y. Zhang et al. experimented with detecting mental fatigue through EEG signals to reduce the probability of traffic accidents. Their research work is based upon logistic regression, one-way analysis of variance, and recursive feature elimination (logistic-ARFE), and they implemented Gaussian SVM for classification purposes and achieved an accuracy of 73.33% [32]. In recent research, Chen, J. et al. developed a BP neural network model for detecting fatigue through eye closure, yawning, and percentage of eye closure time (PERCLOS) from the recorded video. The accuracy of the proposed model with facial expressions increased by 8.4%. The proposed model can filter out artifacts caused due to facial expressions while detecting driver fatigue [33]. Furthermore, Li, Y. et al. proposed a lightweight wearable device based upon a convolution neural network for detecting driver fatigue through eye images [34]. As fatigue detection technologies continue to advance, they have the potential to improve safety and performance. With the development of technologies and machine learning algorithms, it is becoming easier to detect fatigue. Still, there is a requirement to overcome the research gaps in this field.

## 2. Research Gaps

Researchers are making many efforts to detect stress through various modalities. Nevertheless, more research is needed to develop more reliable methods for detecting fatigue in real-world settings. There is a scope for improvement to enhance the detection accuracy and interface medium. After studying the available literature, the following research gaps have been identified.

The existing systems to detect fatigue due to the lack of sleep are primarily intrusive. In such kinds of systems, several biomedical sensors must be employed on an ambulatory subject’s body. The main disadvantage of using intrusive methods is that the subject remains aware during the detection process. Thus, there is a chance of a biased result in that case. Moreover, intrusive systems restrict the subject’s movement, leading to distortion.Furthermore, in most of the previous studies, uni-modal systems have been considered for such purposes, i.e., features from one domain are implemented. There is a possibility that the outcome will be significantly more accurate by using a multimodal feature fusion approach and hence include more features/parameters. Therefore, a nonintrusive system based on multimodal feature fusion is required to better assess fatigue caused by inadequate sleep.

## 3. Material and Methodology

### 3.1. Material

The multimodal fatigue detection system’s apparatus consists of a desktop computer of Make: HP, Model: R-15 (including wireless keyboard and mouse) used as an interface for signal processing and computation task. A high definition (HD) webcam of Make: Logitech, Model: C920, captures visual spectrum images. Flir’s C2 thermal camera captures thermal images, as shown in Figure 1. A MATLAB version: 2016a software, incorporated with the image processing toolbox, is used for multimodal information processing and categorization.

### 3.2. Subjects

A group of 120 healthy subjects was constituted between 20 and 28 years. They also abstained from caffeine, alcohol consumption, and psychoactive drugs that may influence sleep patterns. All subjects were well informed about the study’s purpose, and we instructed them to fill out a subjective sleepiness scoring questionnaire. Subjects were kept awake for 18 h and called for data acquisition in a controlled environment. When they had taken sound sleep for more than 8 h, the same subjects called again for the baseline (alert state) data acquisition. For consistent circadian rhythm, data was acquired in the morning between 7:00 A.M. to 9:00 A.M. Data acquisition comprises four domains as follows: 1. Visual facial image capturing. 2. Thermal image capturing. 3. Voice acquisition. 4. Keystroke data acquisition. Data acquired from the mentioned four domains are required for cumulative dataset preparation. This dataset is used for developing, testing, and validating the classifier. The entire procedural process is discussed in Section 3.3.

### 3.3. Methodology 

Human emotions can be distinguished through facial expressions, gestures, speech patterns, and physiological indicators [35]. L. Tang et al. proposed an image fusion framework for infrared and visible images in real time [36]. This experimental study aims to identify the significant physiological parameters from different domains for fatigue detection and develop a computer-aided system using multimodal features. 

Due to sleep deprivation, there will be a considerable change in the various physiological parameters. The significant parameters for detecting human fatigue are eye cues, facial expressions, body reflexes, voice patterns, and the facial region’s thermal spectrum. An experimental apparatus has been prepared to detect these physiological parameters and behavioral changes, as described in Section 3.2. A graphical user interface has been developed for interface purposes and to facilitate the whole process.

The overall procedure is carried out in five stages, as depicted in Figure 2. This apparatus designed by us is a multimodal feature fusion from four domains as follows: 1. Visual spectra facial image analysis, 2. Keystroke analysis, 3. Voice analysis, and 4. Thermal spectra image analysis. The whole methodology consists of five stages, including 1. Data acquisition, 2. Feature extraction, 3. Multimodal feature fusion, 4. Processing and analysis, and 5. Classification and detection.

#### 3.3.1. Data Acquisition 

The foremost step is to acquire data in a noise-free and constrained environment. A protocol has been made to keep the dataset uniform and get reliable results, and the apparatus has been installed in an isolated chamber with diffused lights for consistency in ambient light. A direct light source was not there in the chamber, reflecting the subject’s face. Further, we maintained consistency in ambient temperature, which was within the range of 22 °C to 28 °C. External sound interference was negligible. The subject’s seating distance was 50 to 60 cm apart from the apparatus.

#### 3.3.2. Feature Extraction 

Significant features have been obtained from the four domains to detect fatigue, i.e., a. Visual image features, b. Thermal image features, c. Keystroke dynamics, and d. Vocal features.

(a) Visual Spectra Facial Image Features: We obtain discrete pictures from the webcam, and the facial region is confined automatically from the discrete images using the Viola-Jones algorithm. Afterward, the Viola-Jones algorithm detects the eye-pair region automatically in the particular facial region. The eye-pair region is cropped and converted into a grayscale image for processing time minimization. Here, cropping and converting RGB images to grayscale images have improved overall efficiency. In the subsequent step, histogram equalization is performed, followed by morphological operations.
(1)$$S_{N}=T\left(r_{N}\right)=\sum_{j=0}^{N}P_{r}\left(r_{j}\right)=\sum_{j=0}^{N}\left(\frac{nj}{n}\right)$$

Here, *SN* represents the intensity value of the resulting image, corresponding to the intensity amount of an input image *rN*; *N* represents total grayscale levels, and *P* represents probability. Morphological operations incorporate four steps; the first step is the disk structure element per retina’s shape. The second and third stages comprise the opening of the image function, trailed by the closing of the image closing function to form a filter. The filtered image is processed through a dilation operation to get notable edges from the dilated image obtained in the fourth step. After morphological operations, the image is converted into binary, later utilized for the decision-making process. We used the image-profile function on a binary image segment to differentiate between closed and open eye pairs. The image profile function provided us with notable information through the pixel value plot of a line. We decide on a threshold limit based on pixel values. Hence, a binary classifier is formed to classify open and closed eyes.

Open and closed eyes are differentiated by a binary algorithm developed for classification. Eight perpendicular lines are formed when an eye pair is open across the image profile line. These perpendicular lines characterize the number of white pixels across the lines. The opened eyes, nose, and pupil segments make more than eight perpendicular lines. Whereas in closed eyes, only nose boundaries are observed, resulting in only two perpendicular lines in a function of the image profile. This binary algorithm can automatically distinguish open or closed eye pairs across the perpendicular line count. Afterward, significant features were extracted from the images, followed by feature reduction. Feature reduction is performed through Fisher’s discrimination ratio (FDR). Elevated FDR denotes that the feature’s capability to discriminate between two classes is higher [37]. The FDR value for the nth feature is calculated as described in Equation (2).
(2)$$FDR_{n}=\left|\frac{\mu_{an}-\mu_{bn}}{\sqrt{\left(\sigma_{an}^{2}+\sigma_{bn}^{2}\right)}}\right|\;\forall \;n\in \left[1,11\right]$$

Here, *µa* = mean of class ‘*a*’; *µb* = mean of class ‘*b*’; *σa* = standard deviation of class ‘*a*’ and *σb* = standard deviation of class ‘*a*’. Based on the highest FDR and minimum correlation coefficient, the six most significant features have been identified out of ten facial image features, as shown in Table 1. 

(b) Thermal Image Features: Overall, five features (listed in Table 2) are considered significant from the captured thermal images. The facial region is initially cropped and extracted from the image acquired from the thermal camera to extract the pixel count value. Then, the extracted thermal image of the facial region is converted into grayscale. Cropping the image and converting it to grayscale reduces the processing time significantly. Afterward, the grayscale image is transformed into a binary image, as shown in Figure 3.

To extract the other four thermal features (two from the periorbital region and two from the forehead region), the ROIs from the thermal image have been selected, as shown in Figure 3. During experiments, it has been observed that the periorbital region and forehead region are significant RIOs. Thermal and visual spectrum images are captured by the same camera, i.e., Flir C2. The visible spectra image is resized to match the thermal image size, i.e., 320 × 240 pixels. Afterward, the eye pair region is extracted from the visual spectrum image using the Viola-Jones algorithm, and we get a bounded box over the eye pair. The coordinates of the bounded box to get the ROIs for the periorbital region from the thermal image. The exact size bounded box is put adjoining above this to obtain forehead features. After getting ROIs, the next step is to obtain FPS (Fourier power spectrum) features from selected ROIs. This texture model provides texture features related to information, like contrast, grain size, and orientation. The discrete Fourier transform (DFT) approach has been used to quantify the texture features. Afterward, the angular sum and the radial sum of the DFT were calculated to obtain the texture features. FPS features are computed through the frequency domain and power spectrum.
(3)$${\left|F\left(u,v\right)\right|}^{2}=F\left(u,v\right)\;F\times \left(u,v\right)$$

Here, the DFT of an image is *F* (*u*, *v*), and *F* × (*u*, *v*) is the complex conjugate of the DFT of an image. To obtain function *S*_r_(*θ*), spectral features are represented as polar coordinates. Further, frequency is defined as *S*_r_(*θ*), and direction is expressed as *S*_*θ*_(*r*). Wedge analysis is performed by evaluating *S*_*θ*_(*r*) for a fixed value of *θ*. It is the spectrum behavior along the radial direction from an origin. In contrast, the ring analysis is performed by evaluating *S*_r_(*θ*) for a fixed value of *r*. It is the behavior of the entire range beside a concentric circle on the origin. The summation of these discrete values gives the global interpretation:(4)$$S_{\theta }=\sum_{\theta =0}^{\pi }S_{\theta }\left(r\right)$$
(5)$$S_{r}=\sum_{\theta =0}^{R_{0}}S_{r}\left(\theta \right)$$
where *R*_0_ is the radius of the concentric circle on the source. In the current texture model, two features: *S*_r_ and *S*_*θ*_ are computed by determining the texture’s orientation.

(c) Keystroke Dynamics: It is a behavioral biometric; used to observe the keyboard actions when the subject types a sentence through a keyboard. Through experiments, the variation in the keystroke attributes for fatigued subjects can be distinguished from the alert subjects. Keystroke dynamics measure the user’s typing pattern when an individual subject types on the keyboard. The features from keystroke dynamics are listed in Table 3.

A total of four features have been identified to measure keystroke dynamics. The first feature is the keystroke rate. It is the total number of keypresses by an individual to complete one string, as shown in Figure 4.

Different sentences of similar character length have been used in the experiment, i.e., around 48 to 54 characters, considering space keypress. The purpose of distinct sentences is that the individual does not become habitual to typing one sentence every time. The second feature is the difference in characters; this feature is measured by calculating the total number of mismatches between the string displayed on the screen and the string entered by the subject using a keyboard. Each character is compared one by one with the preliminary assumption that the subject is prone to make mistakes if the subject is exhausted. In contrast, the errors attempted by an active subject will be negligible. The third feature is the ASCII difference; here, the preliminary assumption is that sometimes the subject is alert, but characters’ sequence changes may occur while typing a sentence in a hurry. There will be no difference in ASCII values of that particular sentence in such cases. The fourth feature is the total time taken by an individual to enter a sentence or string, i.e., whole string time, as shown in Figure 5. Let us take a string, “With the new day comes new strength and new thoughts”, as an example. This particular string consists of 52 characters, including the space keypress. If the subject is fatigued, he will take more time within the keypress. Further, there are chances he will make some spelling mistakes. Suppose the subject is alert and makes spelling mistakes, which can be detected by the characters’ differences in the ASCII value. If the subject is alert, there will be no ASCII difference. Further, the fatigued subject takes more time to complete all the keypresses than the alert subjects.

(d) Vocal Features: Five significant features are extracted from the subjects’ voice samples to detect fatigue, as listed in Table 4. The first feature is the fundamental frequency or pitch of auditory sensation produced by the speech. The second feature is the rate of speech, also known as voiced, and unvoiced duration represents temporal speech rhythm characteristics, such as pause patterns. The third feature is the sound pressure level, which is the difference between the pressure of the sound wave produced and the ambient pressure. The fourth feature is power spectral density; it is a limited average power of a speech signal described by the average power spectral density.

Power spectral density is the area under its Fourier transform magnitude. The average power of a signal s(t) is *Pavg*; computed as in Equation (6).
(6)$$\underset{T\to \infty }{P_{avg}=\lim}\frac{1}{2T}\int_{-T}^{T}s{\left(t\right)}^{2}\mathrm{dt}$$

Speech duration is the fifth feature, representing the time taken by a subject to complete a sentence fragment. A sample voice spectrum for 10 s audio of the active subject and the fatigued subject is shown in Figure 6.

#### 3.3.3. Proposed Multimodal Feature Fusion

Feature fusion is very effective when more than two domains are involved. In this study, four different domains have been studied. The need for multimodal feature fusion arises to obtain the optimum results. During the experiments, it has been observed that accuracy could be enhanced by combining the features from multiple domains, i.e., feature fusion. Thus, ten combinations have been evaluated to improve the results, e.g., by combining features from two domains and then from three domains and assigning different weights to each domain. The results obtained from feature fusion with various combinations are shown in Table 5.

In this multimodal feature fusion approach, the critical part is predictive modeling through inductive reasoning that involves past evidence to determine the outcome and keeps on adding the dataset after every trial. This is achieved by developing adaptive models trained on past data and making predictions on new data. We took 120 samples (60 alert + 60 fatigued) from two classes from both classes. Initially, we trained our model with 80 samples (40 alert and 40 fatigued), tested on the remaining 40 samples (20 alert + 20 fatigued), and performed cross-validation by using five different classifiers for a baseline of performance. It is explicitly stated that the testing in each fold is done on the data which is not exposed to the classifier during its training.

Furthermore, we decided to combine the features from all four domains to improve detection accuracy. In the beginning, features from the four domains were combined, each with 25% weight. After some iterations, we implemented the optimization technique to assign the empirical weights (*α*, *β*, *γ*, and *δ*) to the respective domain with the goal of achieving maximum classification accuracy. After acquiring features from all domains, features are fused through an adaptive framework for autonomous weighting. The proposed feature fusion system allows the feature-level fusion of signals from multiple domains. Weights are assigned automatically after optimization through the MATLAB optimization toolbox using the multiobjective optimization goal attainment method. Here, *S*1 + *S*2 + … *Sn* represents *n* number of subjects with weights respective to each domain, i.e., *α* for visual spectrum images *β* for thermal images, *γ* for keystroke, and *δ* for vocal features. The cumulative result was obtained as a summation of all the empirical weights of respective subjects, as explained in Equation (7).
(7)$$\sum \;\left(\alpha 1+\beta 1+\gamma 1+\delta 1\right)S1+\left(\alpha 2+\beta 2+\gamma 2+\delta 2\right)S2+\cdots +\left(\alpha n+\beta n+\gamma n+\delta n\right)Sn$$

Finally, the features from all four domains are fused to get optimized results, as shown in Figure 7.

#### 3.3.4. Classification

Empirical weights have been assigned to the features to improve the classifier’s accuracy. Initially, we assigned 25% weight to all the domains. Afterward, we applied the optimization technique to assign the weights to achieve maximum classification accuracy. We assigned 29% for visual spectrum image analysis, 37% for thermal spectrum image analysis, 16% for keystroke analysis, and 18% for speech analysis. For this proposed multimodal feature fusion technique, a threshold value is taken as 0.5. It implies that if the output of the proposed multimodal feature fusion technique is less than 0.5, then the subject is alert; otherwise, the subject is classified as fatigued.

To optimize the classification through multimodal feature fusion, 80 samples were taken for training (40 fatigued + 40 alert). These were executed through five different classifiers to assess classification accuracy. The first classifier was k-nearest neighbors (kNN); it provides an accuracy of 85% with 34 correctly classified instances out of 40. The second classifier is random tree, in which an accuracy of 87.5% has been achieved with 35 correctly classified instances out of 40. In comparison, 35 cases are correctly classified through random forest with an accuracy of 87.5%. Support vector machines (SVM) and multilayer perceptron gave an accuracy of 90% by detecting 36 correctly classified out of 40 as shown in Table 6. The highest accuracy of 92.5% was obtained by the proposed method detecting 37 correctly classified subjects out of 40. This was subjected to three-fold cross-validation giving an average accuracy of 93.33%.

## 4. Results and Discussion

If we discuss individual domain-wise fatigue detection in the first domain, i.e., visual spectra image analysis, fatigue detection accuracy is 67.5%. This is given in Table 7. In the thermal domain, an upsurge is observed in the detection accuracy. i.e., thermal images, with an accuracy of 75%. However, there is a decline in detection accuracy in the keystroke domain, i.e., keystroke analysis, with an accuracy of 67.5%. An accuracy of 70% has been observed through the vocal domain. Finally, an accuracy of 92.5% is achieved through the multimodal feature fusion technique. This being the highest accuracy of all the techniques tested, is subjected to three-fold cross-validation. 

## 5. Conclusions and Summary

There will be a detrimental effect on the individual’s physical performance and capabilities due to poor sleep. Fatigue induced by sleep deprivation is a significant cause of accidents because fatigue affects awareness and worsens drivers’ and operators’ vigilance [38]. Further, fatigue assessment is challenging since the clinical factors for the quantification of drowsiness are not well defined. The proposed methodology for fatigue detection is based on feature fusion techniques.

The experiment demonstrates that the performance is optimized after feature fusion from all four domains. The empirical weights have been assigned during feature fusion according to the detection accuracy for classifier optimization.

The confusion matrix for the proposed multimodal feature fusion technique is shown in Table 8. Individually, the thermal domain has a maximum detection accuracy of 75%. The second highest is the vocal analysis, with a detection accuracy of 70%. The visual spectra and keystroke dynamics are on the third rank, with a detection accuracy of 67.5%. To conclude, the proposed multimodal feature fusion technique exhibits the highest detection average accuracy of 93.33% in three-fold cross-validation.

## Figures and Tables

**Figure 1 sensors-23-04129-f001:**
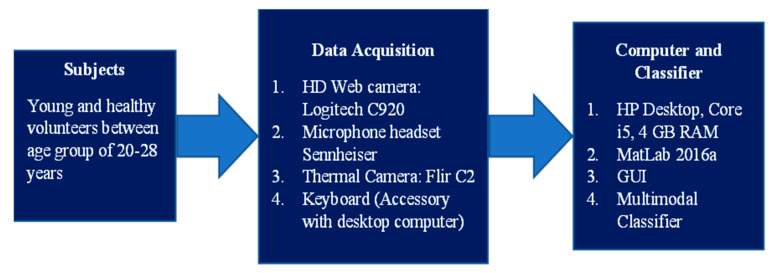
Demonstrating experimental apparatus.

**Figure 2 sensors-23-04129-f002:**
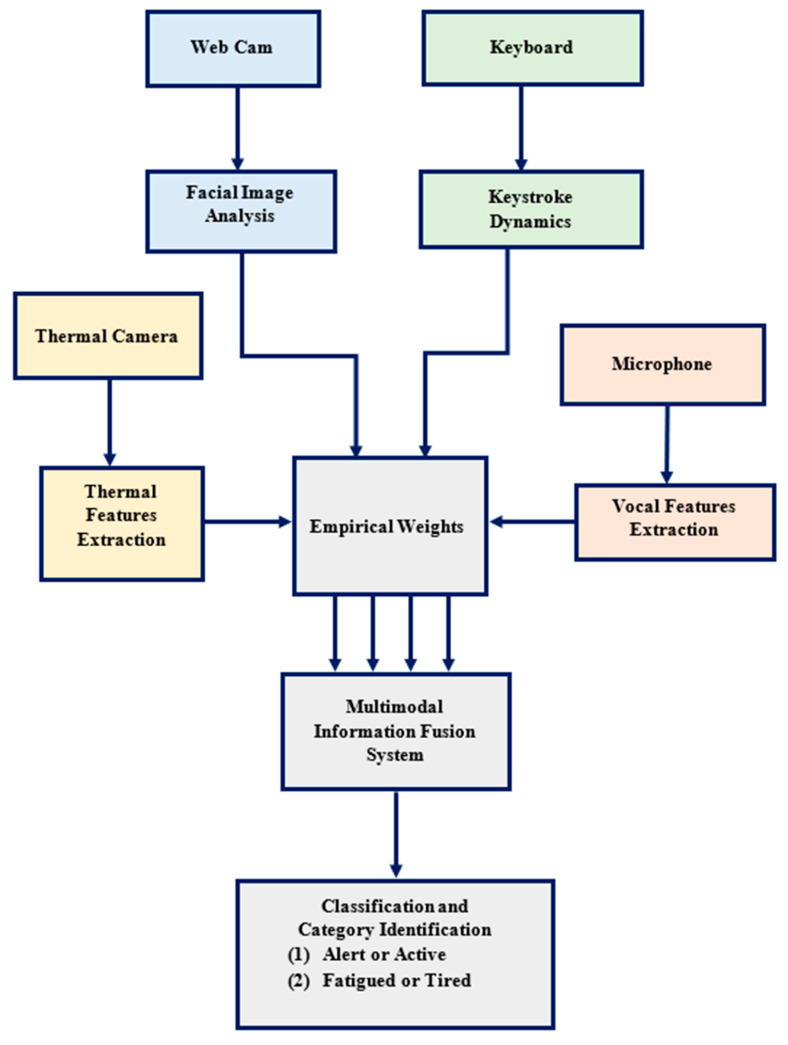
Exhibits methodology for fatigue detection using multimodal fusion.

**Figure 3 sensors-23-04129-f003:**
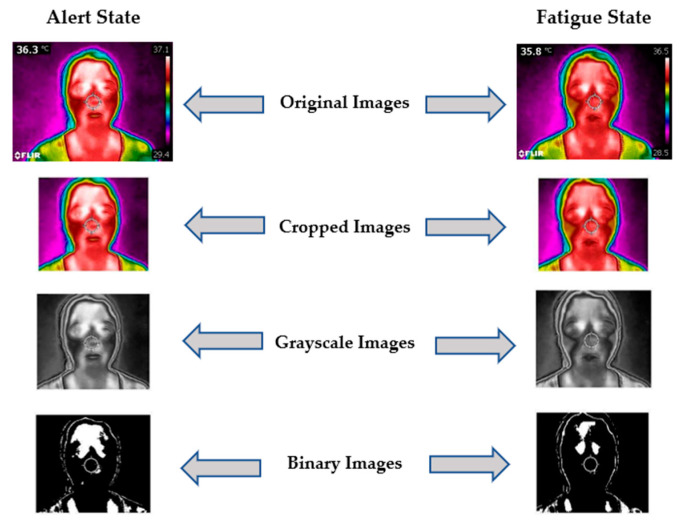
Depiction of thermal image processing.

**Figure 4 sensors-23-04129-f004:**
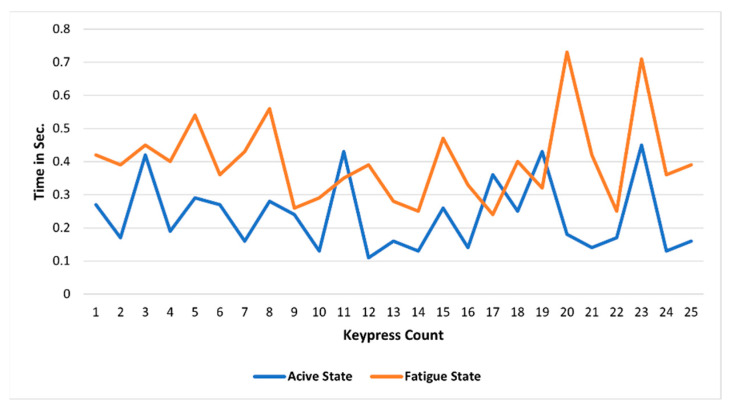
Active vs. fatigue keystroke pattern for the same word and same subject.

**Figure 5 sensors-23-04129-f005:**
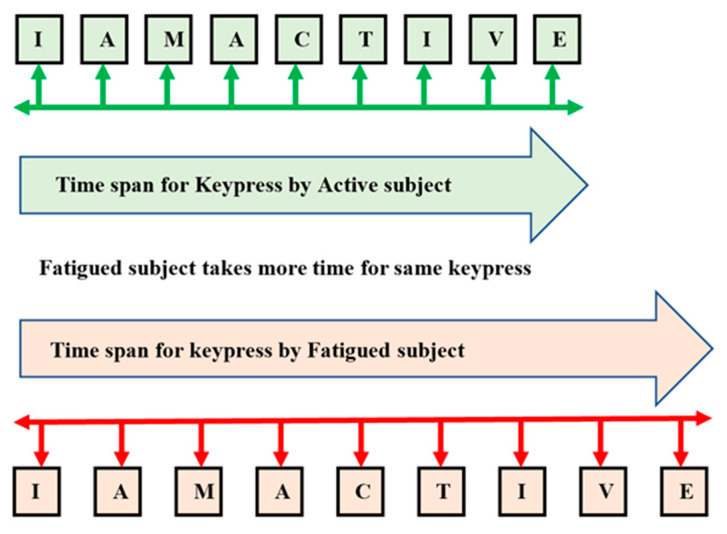
Active vs. fatigue keypress count.

**Figure 6 sensors-23-04129-f006:**
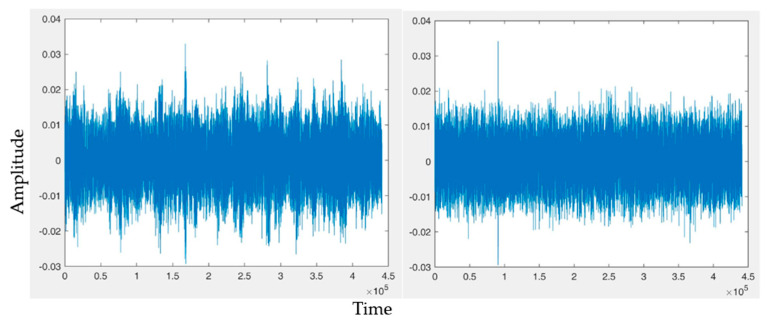
Active vs. fatigue voice pattern.

**Figure 7 sensors-23-04129-f007:**
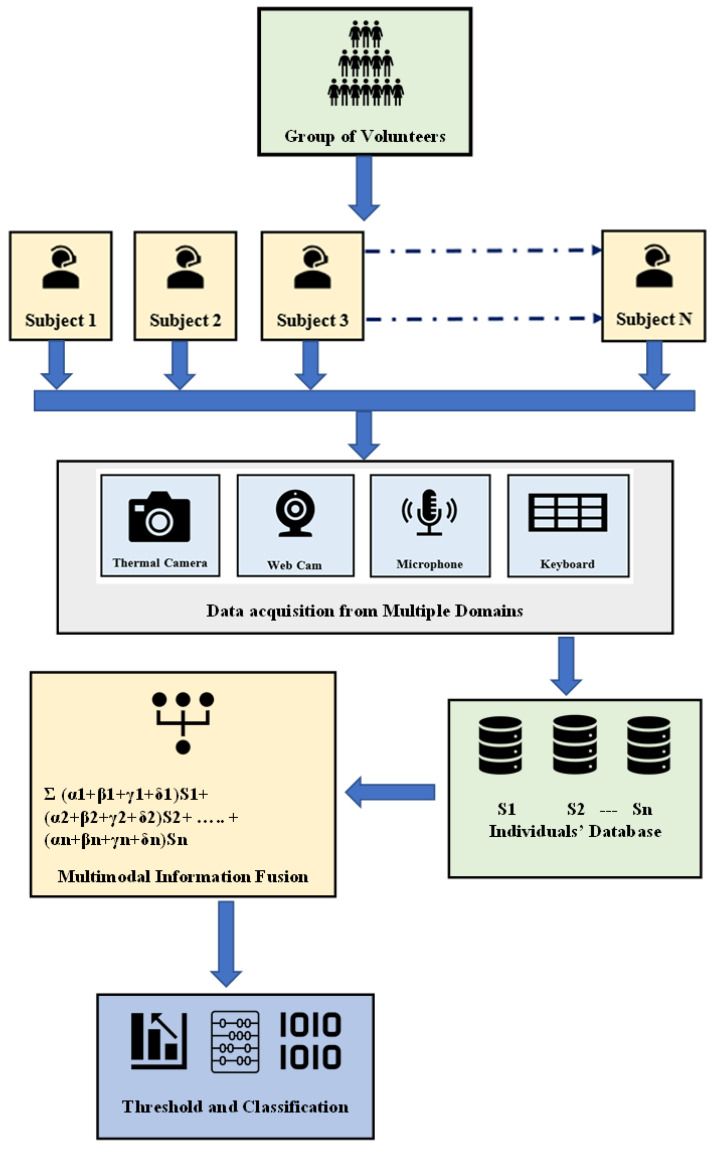
Exhibits methodology for fatigue detection using multimodal fusion.

**Table 1 sensors-23-04129-t001:** List of features from visual spectra facial image.

S. No.	Features	Description
1	Eyes Close Period (ECP)	The time period for closed eyes in a 3-min video sample
2	Open to Close Changeover Period (OCOP)	The time for the eye state to change from an open state to the closed state
3	Close to Open Changeover Period (COCP)	The time for the eye state to change from a closed state to an open state
4	Total Changeover Time (TCT)	Time period for total changeover states (OCOP+COCP)
5	Inter Changeover Frame Count (ICFC)	Changeover frames count
6	Eyes Blink Count (BKC)	Total eye blinks number in a 3-min video sample

**Table 2 sensors-23-04129-t002:** List of features from the thermal image.

S. No.	Features	Description
1	Pixel count	Total number of white pixels in binary image
2	FHFa	Angular Sum of Forehead region
3	FHFr	Radial sum of forehead region
4	PRFa	Angular sum of periorbital region
5	PRFr	Radial sum of periorbital region

**Table 3 sensors-23-04129-t003:** List of features from keystroke dynamics.

S. No.	Features	Description
1	KSR	Keystroke Rate
2	CRE	Character Error
3	AVD	ASCII Value Difference
4	TST	Total String Time

**Table 4 sensors-23-04129-t004:** List of vocal features.

S. No.	Features	Description
1	FFP	Fundamental Frequency (Pitch)
2	RSH	Rate of Speech
3	SPL	Sound Pressure Level
4	PSD	Power Spectral Density
5	SPD	Speech Duration

**Table 5 sensors-23-04129-t005:** Detection accuracy matrix from the individual domains.

S. No.	Domains	Equal Empirical Weights	Accuracy	Optimized Empirical Weights	Optimized Accuracy
1	Visual + Thermal	0.5 + 0.5	75%	0.4 + 0.6	77.5%
2	Visual + Keystroke	0.5 + 0.5	70%	0.65 + 0.35	75%
3	Visual + Voice	0.5 + 0.5	70%	0.62 + 0.38	72.5%
4	Thermal + Keystroke	0.5 + 0.5	72.5%	0.7 + 0.3	75%
5	Thermal + Voice	0.5 + 0.5	72.5%	0.55 + 0.45	77.5%
6	Keystroke + Voice	0.5 + 0.5	70%	0.31 + 0.69	72.5%
7	Visual + Thermal+ Keystroke	0.34 + 0.33 + 0.33	75%	0.36 + 0.38 + 0.26	77.5%
8	Visual + Thermal + Voice	0.34 + 0.33 + 0.33	82.5%	0.34 + 0.36 + 0.30	87.5%
9	Thermal+ Keystroke + Voice	0.34 + 0.33 + 0.33	80%	0.35 + 0.25 + 0.30	85%
10	Visual + Thermal + Keystroke + Voice	0.25 + 0.25 + 0.25 + 0.25	82.5%	0.29 + 0.37 + 0.16 + 0.18	92.5%

**Table 6 sensors-23-04129-t006:** Multimodal feature fusion classification accuracy.

S. No.	Name of Classifier	Accuracy %	Correctly Classified Instances	Incorrectly Classified Instances
1	kNN	85	34	6
2	Random Tree	87.5	35	5
3	Random Forest	87.5	35	5
4	SVM	90	36	4
5	Multilayer Perceptron	90	36	4
6	Proposed Method	92.5	37	3

**Table 7 sensors-23-04129-t007:** Detection accuracy matrix from the individual and multimodal domains.

Domains	Subject	True	False	Accuracy
1. Visual spectra image	Positive	13	7	67.5%
	Negative	6	14	
2. Thermal spectra image	Positive	16	4	75%
	Negative	6	14	
3. Keystroke dynamics	Positive	14	6	67.5%
	Negative	7	13	
4. Vocal Analysis	Positive	14	6	70%
	Negative	6	14	
5. Proposed Multimodal feature fusion technique	Positive	18	2	92.5%
	Negative	1	19	

**Table 8 sensors-23-04129-t008:** Confusion matrix.

Domain	Folds		True	False	Accuracy
Multimodal Feature Fusion	1	Positive	19	1	92.5%
		Negative	2	18	
	2	Positive	19	1	95%
		Negative	1	19	
	3	Positive	18	2	92.5%
		Negative	1	19	
	Average Accuracy				93.33%

## Data Availability

Data that support the findings of this study are available from the corresponding author upon a reasonable request. Sharing of images is restricted due to privacy concerns.

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
