# Peer review of "A Multimodal Feature Fusion Framework for Sleep-Deprived Fatigue Detection to Prevent Accidents"

_sensors, 2023, doi:10.3390/s23084129_

Round 1

Reviewer 1 Report

The paper proposes fatigue detection system based on feature fusion from multiple modalities: visual images, thermal images, voice recordings and keystroke recordings.

Research topic is important and appropriate for this journal.

The data acquisition process seems reasonable and sound. It involves 120 subjects (60 fatigued and 60 alert). 

Literature review is extensive, but hard to follow because it is presented as a single paragraph. I suggest dividing it into multiple paragraphs according to some criterion. Maybe group related work based on the same modality...

My main concerns are regarding the results in tables 6 and 7. It seems to me that the tables present results on the training set. The total number of samples in these tables is 120, which is equal to the total number of samples. Therefore, I assume that the same data is used for model training and testing. These results are not valid because they do not reflect the generalization ability of the model. In machine learning, It is usual to show results on the test set (the unseen data during training). 

I am also confused about the necessity of a separate feature fusion method prior the Multilayer Perceptron (MLP).  It seems to me that MLP is able to learn the appropriate feature fusion weights in the first layer through the training process. I encourage the authors to discuss this.

Author Response

Thank you for the feedback, below is my response in Green:

  1. The paper proposes fatigue detection system based on feature fusion from multiple modalities: visual images, thermal images, voice recordings and keystroke recordings. > No action is required.
  2. Research topic is important and appropriate for this journal. > No action is required.
  3. The data acquisition process seems reasonable and sound. It involves 120 subjects (60 fatigued and 60 alert). > No action is required.
  4. Literature review is extensive, but hard to follow because it is presented as a single paragraph. I suggest dividing it into multiple paragraphs according to some criterion. Maybe group related work based on the same modality...  > Section 1.2 is further subdivided into four sections according to the domains for clarity.
  5. My main concerns are regarding the results in tables 6 and 7. It seems to me that the tables present results on the training set. The total number of samples in these tables is 120, which is equal to the total number of samples. Therefore, I assume that the same data is used for model training and testing. These results are not valid because they do not reflect the generalization ability of the model. In machine learning, It is usual to show results on the test set (the unseen data during training). > Thanks for the insightful input. As suggested, we have reworked the process and used 80 samples for training and used remaining 40 samples for testing. This has been done as per 3-fold cross validation. Accordingly, tables 6 and 7 have been modified. More detail has been presented in the section 3.3.4.
  6. I am also confused about the necessity of a separate feature fusion method prior the Multilayer Perceptron (MLP).  It seems to me that MLP is able to learn the appropriate feature fusion weights in the first layer through the training process. I encourage the authors to discuss this. > The reviewer has raised a valid question. We were not able to explain clearly on the use of feature fusion. Actually, the classifiers like SVM and MLP have been used with the most distinguishable features from all the four domains. However, in the proposed method, the weights of the features are sensitive to their discrimination ability. 

Reviewer 2 Report

The main contribution of this paper is to implement the building of a data set for the detection of fatigue caused by sleep deprivation using non-invasive equipments. 

The following problems remain to be addressed:

1. The main differences between this work and other research work are described in the manuscript as follows: (1) modal information extraction is based on non-invasive equipment; (2) Based on an adaptive machine learning method to achieve multi-modal feature fusion. However, the rationale for the implementation of point (2) (the adaptive machine learning approach) is not specified in Section 3.3.3 or elsewhere. 

2. The presentation logic and the content structure of relevant works outlined in Section 1.2 of this manuscript are not clear enough.

3. The naming order of the figures in this manuscript is wrong, from Figure 3 directly to Figure 5 where Figure 4 (Section 3.3.2) is missing. Meanwhile, the citation of each figure in this manuscript is wrong,for example, the citation of Figure 5 is wrong and Figure 4 which did not appear (Section 3.3.b) is mentioned in the texts.

4. The vertical coordinates of the left and right pictures in Fig. 5 (Section 3.3.2) are inconsistent. 

5. The horizontal and vertical coordinates are not indicated in Fig. 7 (Section 3.3.3.b).

6. There are many non-standard symbols in the text. For example, in the third paragraph of Section 3.3.2.b, there are many non-standard symbols. 

7. The confusion matrix mentioned in the conclusion part (Section 5, Table 8) of this manuscript: the confusion matrix is generally used in the test set. For each fold, the corresponding confusion matrix can be drawn, but only the test data of the current fold can be considered. 

8. In the references, the detailed information in [31] is problematic. Secondly, the format of the references is inconsistent in a few references.

Author Response

Thank you for the feedback, below is my response in Green:

  1. The main differences between this work and other research work are described in the manuscript as follows: (1) modal information extraction is based on non-invasive equipment; (2) Based on an adaptive machine learning method to achieve multi-modal feature fusion. However, the rationale for the implementation of point (2) (the adaptive machine learning approach) is not specified in Section 3.3.3 or elsewhere. >  The reviewer has raised a valid question. We were not able to explain clearly on this issue and hence modified the manuscript accordingly.
  2. The presentation logic and the content structure of relevant works outlined in Section 1.2 of this manuscript are not clear enough. > Section 1.2 is further subdivided into four sections according to the domains for clarity.
  3. The naming order of the figures in this manuscript is wrong, from Figure 3 directly to Figure 5 where Figure 4 (Section 3.3.2) is missing. Meanwhile, the citation of each figure in this manuscript is wrong, for example, the citation of Figure 5 is wrong and Figure 4 which did not appear (Section 3.3.b) is mentioned in the texts. > Section 1.2 Naming order corrected, and citation rearranged appropriately.
  4. The vertical coordinates of the left and right pictures in Fig. 5 (Section 3.3.2) are inconsistent. > Graphs in the figure for Active and fatigue state are captured in the same image to eliminate inconsistency due to scaling.
  5. The horizontal and vertical coordinates are not indicated in Fig. 7 (Section 3.3.3.b). > Indicated Amplitude and time on the horizontal axis and vertical axis.
  6. There are many non-standard symbols in the text. For example, in the third paragraph of Section 3.3.2.b, there are many non-standard symbols. > Symbols amended to make them standardized.
  7. The confusion matrix mentioned in the conclusion part (Section 5, Table 8) of this manuscript: the confusion matrix is generally used in the test set. For each fold, the corresponding confusion matrix can be drawn, but only the test data of the current fold can be considered. > Modified table 8 and section 5 for clarity.
  8. In the references, the detailed information in [31] is problematic. Secondly, the format of the references is inconsistent in a few references. > Edited references and citations rearranged.

Round 2

Reviewer 1 Report

The authors have addressed most of my concerns.

Author Response

Thank you for the feedback, below is my response in Green:

  1. The authors have addressed most of my concerns. -> No action is required.
